# Conservative Treatment of Sever’s Disease: A Systematic Review

**DOI:** 10.3390/jcm13051391

**Published:** 2024-02-28

**Authors:** Pablo Hernandez-Lucas, Raquel Leirós-Rodríguez, Jesús García-Liñeira, Helena Diez-Buil

**Affiliations:** 1Functional Biology and Health Sciences Department, University of Vigo, 36001 Pontevedra, Spain; 2SALBIS Research Group, Nursing and Physical Therapy Department, University of Leon, 24401 Ponferrada, Spain; 3Special Didactics Department, University of Vigo, 36001 Pontevedra, Spain; jesgarcia@alumnos.uvigo.es; 4Faculty of Physiotherapy, Gimbernat University Schools, 08172 Sant Cugat del Valles, Spain

**Keywords:** calcaneal apophysitis, rehabilitation, sports disease, children, heel, physical therapy modalities, physical therapy specialty

## Abstract

**Background**: Sever’s disease, or calcaneal apophysitis, stands as the most prevalent cause of heel pain in children, often linked to sports like soccer, Australian football, and basketball. While various therapies are documented in the scientific literature, the standard choice is conservative treatment. Thus, the objective of this research was to assess the effectiveness of diverse conservative methods and techniques in alleviating Sever’s disease symptoms. **Methods:** Systematic searches were conducted in October 2023 in PubMed, Web of Science, Scopus, SportDiscus, and PEDro, using terms like Osteochondritis, Osteochondrosis, Apophysitis, Sever’s disease, Calcaneus, Adolescent, Child, and Childhood. The PEDro scale gauged methodological quality, and the Cochrane Risk of Bias tool evaluated the risk of bias. **Results**: Eight randomized controlled studies were included, featuring commonly used treatments such as insoles, therapeutic exercises, Kinesio taping, and foot orthoses. The methodological quality was generally good, with an average PEDro score of 6.75 points. Regarding bias, four articles had low risk, three had high risk, and one had some concern. **Conclusions:** Conservative treatment emerges as an effective option for alleviating symptoms associated with Sever’s disease.

## 1. Introduction

Sever’s disease (SVD), also known as calcaneal apophysitis, was initially described by James Warren Sever in 1912 [1,2,3]. It is described as an overuse injury with an insidious onset of pain and is usually not related to a traumatic event [4]. Histologically, it is defined as inflammation or bone edema resulting from a stress fracture in the secondary growth center of the calcaneus [2]. This inflammation arises due to repetitive traction forces from the triceps surae muscle on the calcaneal apophysis [2]. In addition, with X-Ray, it is possible to see an increase in the density of the epiphysis [3].

Sever’s disease stands as the most common cause of heel pain in children and young individuals [1,2,5,6,7], with an incidence of 3.7 per 1000 patients [1,2,5]. It is a common musculoskeletal disorder, representing between 2% and 16% of consultations in sports clinics [8,9] and accounting for 5.8–22.7% of repetitive stress injuries in children [10,11,12]. During the process of skeletal development, a remarkable fact is observed when the calcaneal process remains in an open state [7,13], with more prevalence observed in girls between 7 and 9 years and boys between 8 and 15 years, peaking between 10 and 12 years of age [2,14]. Pain tends to be bilateral and asymmetrical [14]. Full recovery of the SVD is expected after closure of the calcaneal apophysis [6,7,15].

Limited ankle dorsiflexion is a common intrinsic factor, while extrinsic factors include alterations in foot alignment [2]. There is inconsistency in the scientific literature regarding body mass index as a risk factor, with some studies not identifying it as significant [16,17] while others do [4,5]. Nevertheless, asymmetry has been observed in the distribution of weight in patients with SVD, being greater in the heel with more pain [18]. Engagement in high-intensity sports like soccer, Australian football, and basketball is proposed to contribute to SVD development [17,19]; also, enrolling in a new sport may be associated with it [6]. Additionally, cartilaginous diseases such as Osgood–Schlatter disease have been observed to play a role in the development of SVD [20].

Currently, there are no clinical guidelines recommending the best treatment for SVD, but previous systematic reviews have emphasized the goal of facilitating tissue healing [1]. The first goal of treatment is pain relief [6]. A proper clinical investigation is important in obtaining this goal, including history-taking and a physical exam [21]. Usually, a positive heel squeeze test is sufficient and indicative of SVD. Nevertheless, radiographic imaging can exclude other differential diagnoses [4,22,23].

Commonly used treatments include stretching and lengthening of the triceps surae muscle, ice application, restriction or limitation of physical activity, rest, topical non-steroidal anti-inflammatory drugs, taping, and the use of foot orthoses and heel cushions [1,2,4,24]. Foot orthoses and heel cushions are often considered the most effective methods for reducing pain scores [25]. Patients tent to be pain-free after a time period ranging from a few weeks to several months [4,7]. Return to play fluctuates from two to eight weeks [4,7,24]. It is important to educate parents and coaches for the early diagnosis of SVD [1]. Although there are no clinical guidelines for preventing SVD, systematic reviews highlight the importance of educational, environmental, and physical interventions [1].

A systematic review published in 2013 focused on interventions for pain reduction in SVD, including only three randomized controlled trials [26]. Currently, there is no updated systematic review analyzing the effect of conservative treatment in SVD, despite it being standard care [1]. Therefore, the aim of this systematic review was to analyze and evaluate the scientific evidence of the effects of different conservative methods and techniques to improve symptoms of SVD.

## 2. Materials and Methods

### 2.1. Data Sources and Searches

This study was prospectively registered on PROSPERO on 30 September 2023, with the code CRD42023464493. The research adhered to the Preferred Reporting Items for Systematic Reviews and Meta-analyses (PRISMA) guidelines [27]; Exercise, Rehabilitation, Sport Medicine, and Sports (PERSIST) recommendations [28]; and the Cochrane Collaboration guidelines [29,30]. The PICO question was formulated as follows: P—population: patients with SVD; I—intervention: conservative methods and techniques; C—control: surgical treatment, placebo, or non-intervention; O—outcome: characteristics of pain and disability; S—study designs: randomized controlled trials.

A systematic search of publications was conducted in October 2023 across multiple databases, including PubMed, Web of Science, Scopus, SportDiscus, and PeDro. The search strategy involved several combinations, using Medical Subject Headings (MeSH) terms such as Osteochondritis, Osteochondrosis, Calcaneus, Adolescent, and Child. Additionally, free terms like Apophysitis, Sever’s disease, Sever, Calcaneus, and Childhood were included. The detailed search strategy used, according to the focused PICOS question, is presented in Table 1.

### 2.2. Study Selection

After removing duplicates, two reviewers (P.H.-L.; H.D.-B.) independently conducted the initial screening of articles for eligibility. In instances of disagreement, a third reviewer (R.L.-R.) was involved in making the final decision on whether to include the study or not. The following inclusion criteria were applied for study selection: (i) articles employing conservative techniques (e.g., insoles, therapeutic exercise, taping, and foot orthoses, etc.) and (ii) studies with samples consisting of participants with SVD. Studies were excluded if they met the following criteria: (i) studies other than randomized controlled trials and (ii) research involving participants older than 19 years.

Following the initial screening, titles and abstracts were screened for inclusion criteria, and the selected abstracts were obtained in full text. Titles and abstracts lacking sufficient information regarding inclusion criteria were also obtained as full texts. Full-text articles were selected if they met the inclusion criteria, with both reviewers using a data extraction form. The two reviewers (P.H.-L.; H.D.-B.) independently extracted data from the included studies, using a customized data extraction table in Microsoft Excel 2016. In cases of disagreement, the two reviewers engaged in discussions until a consensus was reached.

### 2.3. Data Extraction and Quality Assessment

The following data were extracted for further analysis: demographic information (title, authors, journal, and year), characteristics of the sample (age, gender, inclusion and exclusion criteria, and number of participants), study-specific parameters (duration of the intervention, adverse events, conservative methods, and techniques), and results obtained (variables analyzed, instruments used, and time of follow-up). Tables were used to describe both the characteristics of the studies and the extracted data. The PEDro scale [31] was employed to assess the quality of studies, and the RoB (Risk of Bias) tool [29] was applied to assess the risk of bias. Two reviewers (P.H.-L.; H.D.-B.) conducted the application of the PEDro and RoB scales. In cases of disagreement, a third author (R.L.-R.) was also involved in the process to achieve agreement. Descriptive techniques were used in the quantitative analysis, including the determination of means, standard deviations, and percentages.

## 3. Results

### 3.1. Studies Included

The literature review process identified a total of 411 articles. After removing duplicates and conducting the initial screening, a total of 120 eligible articles were analyzed. After the review, those that matched the object of study of this review and met the inclusion criteria were utilized for analysis; the final analysis was performed on a total of eight studies (Figure 1).

### 3.2. Methodological Quality

The mean methodological quality of all eight articles [26,32,33,34,35,36,37,38] demonstrated good methodological quality [31], with a mean PEDro score of 6.75 points, ranging from 4 to 11 points (Table 2). In terms of bias, four articles were low risk [35,36,37,38], three were high risk [32,33,34], and one had some concerns [26] (Figure 2).

### 3.3. Participants Characteristics

In the studies analyzed, the participants consisted mostly of male patients. In four of the articles studied [32,33,34,35], the sample was entirely male. In two others, it did not reach 30% of women [37,38]; in one of them, the percentage of females was 90.6% [36], and in just one, the sample was balanced, with 42% females [26]. In relation to age ranges, these ranged from 7 to 15 years old, with mean ages ranging from 10.3 ± 1.6 years [36] to 13.18 ± 2.15 [32]. Subjects were mostly recruited through clinics or hospital services, with medical diagnoses. Similarly, in most of the studies, the participants presented physically active subjects, specifying that they practiced physical activity between 2.7 and 3.7 h per week [38], which was unspecified but used as an inclusion criterion [37]. Also, children were recruited from different sports like soccer [32], high-level athletics [33], or barefoot sports (gymnastics, acrobatics, dance, or martial arts) [36]. Participants in the following four articles [26,36] had no adverse effects, although it should be noted that these were not mentioned in the following articles [32,33,34,35].

### 3.4. Studies Characteristics

In the study design, most studies included specific treatment groups to evaluate the effects of the interventions. Sweeney et al. [36] and Alfaro-Santafé et al. [37] compared the use of heel pads; Kuyucu et al. [32] employed Kinesio taping and a sham version (in addition to considering topical analgesics, stretching, and targeted massage therapy). On the other hand, Wiegerinck et al. [38] employed three treatment groups, including physical therapy, the use of heel raises (ViscoHeel, Bauerfeind), and a wait-and-see approach as a control group, allowing for direct comparisons between interventions. James et al. [26] conducted a study that included four intervention groups with different combinations of footwear and orthoses (usual footwear and heel raise, new footwear and heel raises, usual footwear and orthoses, and new footwear and orthoses), including a “wait and see” control group that allowed for comparisons. In the third study, Perhamre et al. [35] included a comparison of the heel cup and heel wedge. However, they compared the effectiveness of these treatments in relation to the pilot group that did not receive any specific intervention (Table 3).

### 3.5. Outcomes Measurements

In the studies analyzed, the participants mostly consisted of male patients. In four of the articles studied [32,33,34,35], the sample was entirely male. In two others, it did not reach 30% of female [37,38]. In the study design, most studies included specific treatment groups to evaluate the effects of the interventions. Sweeney et al. [36] and Alfaro-Santafé et al. [37] compared the use of heel pads, similarly to Kuyucu et al. [32].

The studies presented their results on the basis of rating scales. Scales for the perception and intensity of pain experienced in the heel such as the Visual Analogue Scale (VAS) were widely used in most studies to measure pain perception in participants [26,32,33,34,35]. This scale provided a basis for evaluating the efficacy of therapeutic interventions in reducing pain. Other studies [36,38] used the OxAFQ (Oxford Ankle Foot Questionnaire) scale to assess foot- and ankle-related function and quality of life.

In the three studies by Perhamre et al. [33,34], the use of the heel cup and heel wedge resulted in a significant reduction in pain, with patients preferring the heel cup. Also, a positive correlation was observed between the level of physical activity and pain reduction in both groups, with a greater reduction in the more intense and more painful activities. In other studies, such as Sweeney et al. [36] and Kuyucu et al. [32], no significant differences in VAS were found between treatment groups, suggesting that the treatments did not significantly influence pain perception.

Regarding the OxAFQ, Sweeney et al. [36] found that the groups treated with Tuli’s Cheetah Heel Cup and Tuli’s The X Brace showed no significant differences in the measurements. The results indicated that the Heel Cup may be effective in reducing pain. On the other hand, the study by Kuyucu et al. [32] used the AOFAS (American Orthopaedic Foot and Ankle Society) Scale to measure ankle dorsiflexion, and, in this analysis, significant differences in dorsiflexion were observed thanks to the use of Kinesio taping compared to the simulated tape, but these differences did not increase over time (Table 3).

### 3.6. Treatments Used and Results for Quality-of-Life Improvement

The treatments in the studies were aligned with the aim of finding therapeutic solutions to improve the quality of life of patients affected by this condition. For this reason, all the studies analyzed and performed measurements using pain perception scales. Sweeney [36], Alfaro-Santafé et al. [37], Kuyucu et al. [32], Wiegerinck et al. [38], James et al. [26], and Perhamre et al. [34] focused on understanding how different treatments affect pain in this population.

There was high variability in the treatments, with a majority examining the use of heel cups and shoe inserts, such as Tuli’s Cheetah Heel Cup and Tuli’s The X Brace [21], custom-made foot orthoses and off-the-shelf heel lifts [37], and the use of heel cups and heel wedges with varying insole thickness, as in Perhamre et al. [33,34,35]. In addition to these interventions, Kuyucu et al. [32] explored Kinesio taping, and Perhamre et al. [34] and Wiegerinck et al. [38] investigated physical therapy. The results supported the effectiveness of these interventions in reducing pain associated with SVD, noting that subjects who used pain reduction mechanisms improved their quality of life (Table 3).

## 4. Discussion

This systematic review focused on evaluating the impacts of conservative treatment in cases of SVD. The findings indicate the beneficial impact of these interventions, particularly highlighting the use of orthopedic insoles [26,33,34,35,36,37,38], physical therapy [38] and Kinesio tape [32]. The application of insoles has shown a reduction in pain and an increase in ankle functionality in patients with SVD [26,33,34,35,36,37,38], although the types of insoles were different, such as a prefabricated heel raising inlay [26,37,38], an elastic surface heel inlay [34,35], a rigid thermoplastic heel inlay [33,34,35], prefabricated orthoses [26], custom-made food orthoses [37], a neoprene ankle brace with a built-in multicell and multilayer waffle heel cup [36], and an elastic foot brace with an integrated silicone strip on the heel strap [36].

The type of orthoses used was very different between studies; therefore, it is difficult to generalize, although those that were personalized provided a better result. This effect may be due to the physiological remodeling necessary for the development of injury-resistant tissues, requiring physical activity with repeated impact on the hindfoot according to Wolff’s Law [39]. In SVD, there seems to be a disturbance in this relationship between stimulus load and developmental capacity [40]. Insoles provide cushioning and therefore reduce the impact force on the calcaneus during heel strike [40]. A previous systematic review, including studies up to May 2012, suggests that the implementation of heel lifts could decrease pain in cases of SVD [26]. However, a cautious interpretation of these results is recommended, as they are based on only three randomized clinical trials [26].

This study reinforces the conclusions of James et al.’s systematic review [26] and additionally reports an improvement in foot functionality in children with SVD after using insoles. This is consistent, considering the close relationship between pain and disability [41]. It is important to note that, although the methodological quality of the studies included in James et al.’s systematic review [26], assessed using the PEDro scale, was rated as poor, this present review classifies the methodological quality of the studies as good [31].

Physical therapy has also proven to be beneficial for heel pain [42] and, specifically, in treating SVD [38]. The studies reviewed highlight the finding that the application of physical therapy, including specific stretches and massages, can lead to a significant decrease in pain and improved ankle functionality in patients with SVD [38]. This therapeutic approach may have a positive impact on recovery by directly addressing ankle dorsiflexion restriction, one of the features associated with SVD [38].

Kinesio tape has also been explored as an intervention in SVD, and Kuyucu et al.’s study [32] suggests it may have benefits in improving ankle dorsiflexion. Although the results of this specific study should be interpreted with caution, as differences did not increase over time, the application of Kinesio tape could be considered as part of a comprehensive treatment strategy to address dorsiflexion limitations and reduce pain in patients with SVD [32].

Overall, these findings support the effectiveness of conservative treatment, including the use of orthopedic insoles, physical therapy, and Kinesio tape, in improving symptoms and functionality in patients with SVD. However, it is essential to consider that, while these results are promising, the heterogeneity of interventions and limited sample sizes in some studies highlight the need for larger and more rigorous future research to consolidate these conclusions. Additionally, it is crucial to consider individual patient characteristics when selecting and customizing conservative interventions for the treatment of SVD.

Another additional explanation for the beneficial effects of insoles [26,33,34,35,36,37,38], as well as physical therapy [38] and Kinesio tape [32], is that SVD may be related to an increase in tension or shortening of the Achilles tendon, a common phenomenon during rapid growth in adolescence [43]. This modification in soft tissue could lead to high traction at the insertion point of the apophysis, causing pain and inflammation [40,44]. Additionally, SVD is associated with other conditions that cause tension in the posterior myofascial chain, such as short hamstring syndrome [45]. Consequently, it has been suggested that a simple heel lift can be effective in relieving this tension when wearing shoes [46,47,48]. Similarly, this tension can be reduced with the help of physical therapy [38,48] or taping [32,44].

Conservative treatment, by relieving pain and improving functionality [26,32,33,34,35,36,37,38], allows children to avoid long periods of physical inactivity. This continuity in physical activity is crucial, as it helps prevent weight gain, which is a significant risk factor for SVD [2,4,16]. Surgery is generally advised against in such cases [8]. Therefore, it is crucial to disseminate and promote conservative treatment alternatives to prevent the need for surgical interventions whenever possible [46].

As there is a systematic review of conservative treatment in heel pain, this shows us other types of conservative treatment with moderate evidence, such as low-level laser therapy, dry-needling of lower leg and foot muscle trigger points, and calcaneal taping, as well as high-quality evidence that ultrasound-guided pulsed radiofrequency achieves heel pain reduction [49]. This may pave the way for new research into conservative treatment in SVD.

SVD is closely related to Osgood–Schlatter disease [50] (the onset of a traction apophysitis as a consequence of repeated contractions of the femoral rectum part of the quadriceps [51]), since one of its risk factors is having SVD [52]. Some treatments of both coincide, since Osgood–Schlatter is also treated with the decrease of physical activity [53,54], the application of cold presses [55], physical therapy [56,57] and stretching [14,56,58,59]. Also, treatment includes a good warm-up before activity and cool-down after it [60], as well as knee orthoses [55].

This systematic review is a pioneering work in analyzing the effects of different conservative treatment methods for SVD. Among the study’s limitations, it is important for the authors to acknowledge the absence of subgroup analyses based on sex and age. Additionally, due to the high heterogeneity of the studies included, it was not possible to conduct a consolidated quantitative analysis of the results through a meta-analysis. Consequently, further research is recommended to compare the effects of various interventions and develop specific treatment protocols for SVD.

## 5. Conclusions

Conservative interventions, such as the use of insoles, physical therapy, and Kinesio taping, seem to have a positive effect in reducing pain and improving functionality in patients with SVD. In addition, the disease seems to affect more men than women. The findings of this research can be valuable for healthcare professionals, enabling them to optimize the effectiveness of their clinical interventions and, consequently, mitigate the significant socio-economic impact that SVD has on children and adolescents.

## Figures and Tables

**Figure 1 jcm-13-01391-f001:**
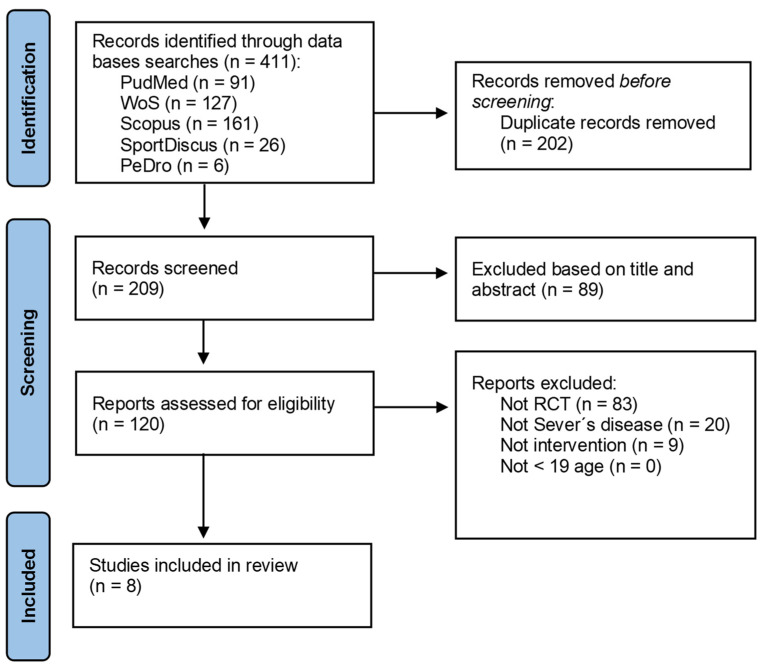
PRISMA flowchart.

**Figure 2 jcm-13-01391-f002:**
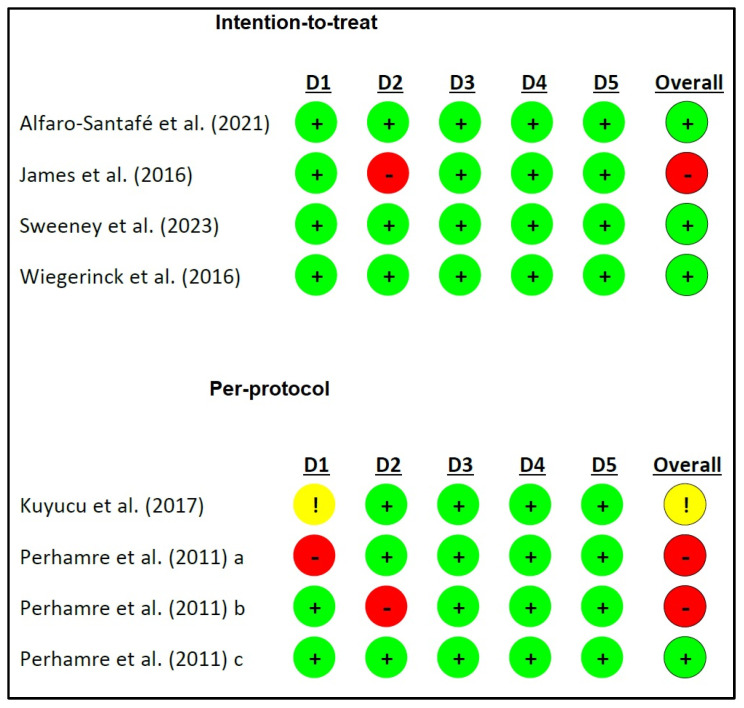
Risk of bias. D1: randomization process; D2: deviations from the intended interventions; D3: missing outcome data; D4: measurement of the outcome; D5: selection of the reported result; !: some concerns; −: high risk; +: low risk. a: reference [35]; b: reference [34]; c: reference [33].

**Table 1 jcm-13-01391-t001:** Search strategy according to the focused question (PICO).

Database	Search Equation
PubMed	(“Osteochondritis” [Mesh] OR “Osteochondrosis” [Mesh] OR “Apophysitis” [TIAB] OR “Sever’s disease” [TIAB] OR “Sever” [TIAB]) AND (“Calcaneus” [Mesh] OR “Calcane” [TIAB]) AND (“Adolescent” [Mesh] OR “Child” [Mesh] OR “Childhood” [TIAB])
Web of Science	TS = ((“Osteochondritis” OR “Osteochondrosis” OR “Apophysitis” OR “Sever’s disease” OR “Sever”) AND (“Calcane”) AND (“Adolescent” OR “Child” OR “Childhood”))
Scopus	TITLE-ABS-KEY (((“Osteochondritis” OR “Osteochondrosis” OR “Apophysitis” OR “Sever’s disease” OR “Sever”) AND (“Calcane”) AND (“Adolescent” OR “Child” OR “Childhood”)))
SportDiscus	TX (“Osteochondritis” OR “Osteochondrosis” OR “Apophysitis” OR “Sever’s disease” OR “Sever”) AND (“Heel bone”) AND (“Adolescent” OR “Children” OR “Youth”)
PeDro	Apophysitis Calcane Child

PICO: P—population, I—intervention, C—control, O—outcome; Mesh: Medical Subject Headings; TIAB: Free text term; TS: Topic; TX: Topic.

**Table 2 jcm-13-01391-t002:** The methodological quality of the included studies.

Author	1 *	2	3	4	5	6	7	8	9	10	11	Score
Alfaro-Santafé et al. (2021) [37]	Yes	Yes	Yes	Yes	Yes	Yes	Yes	Yes	Yes	Yes	Yes	11
James et al. (2016) [26]	Yes	Yes	Yes	Yes	No	No	Yes	Yes	Yes	No	Yes	7
Kuyucu et al. (2017) [32]	No	Yes	No	Yes	No	No	No	Yes	No	Yes	Yes	5
Perhamre et al. (2011) [35]	Yes	Yes	No	No	No	No	No	Yes	No	Yes	Yes	4
Perhamre et al. (2011) [34]	Yes	Yes	Yes	No	No	No	No	Yes	No	Yes	Yes	5
Perhamre et al. (2011) [33]	Yes	Yes	Yes	Yes	No	No	No	Yes	No	Yes	Yes	6
Sweeney et al. (2023) [36]	Yes	Yes	Yes	Yes	No	No	Yes	Yes	Yes	Yes	Yes	8
Wiegerinck et al. (2016) [38]	Yes	Yes	Yes	Yes	No	No	Yes	Yes	Yes	Yes	Yes	8

1: Eligibility criteria specified. * This item relates to external validity and therefore does not contribute to the total score; 2: Subjects randomly allocated to groups; 3: Concealed allocation; 4: Groups were similar at baseline; 5: Blinding of all subjects; 6: Blinding of all therapists; 7: Blinding of all assessors; 8: Measures obtained from more than 85% of subjects allocated to groups; 9: Subjects received treatment or control condition as allocated or intention-to-treat analysis; 10: Between-group statistical comparisons reported for at least one outcome; 11: Both point measures and measures of variability were reported. High, high risk of bias; low, low risk of bias.

**Table 3 jcm-13-01391-t003:** Characteristics of the studies included in this review.

Authors	Sample (% Female)	Intervention	Age (Years)	Period (Months)	Measurements	Results
Alfaro-Santafé et al. (2021) [37]	234 (16.7%)	G1 (*n* = 104): custom-made foot orthoses; G2 (*n* = 104): off-the-shelf heel lifts	11.2 ± 1	3	VAS and algometric threshold; FPI-6 and lunge test; tiltmeter	No differences were observed between groups at baseline in the VAS score, lunge test, and FPI-6. Significant differences between groups were observed in the algometric threshold at baseline. Both groups experienced a significant reduction in pain versus the measurments obtained at baseline. G1 and G2 significantly reduced sports-related pain after treatment. G1and G2 significantly decreased VAS score pain after treatment. G1 and G2 significantly increased the algometric threshold score after treatment. G1 significantly reduced VAS score pain compared to G2 after treatment. G1 resulted in significantly higher levels of improvement, with increased algometry values compared with G2.
James et al. (2016) [26]	124 (42%)	G1 (*n* = 31): usual footwear and heel raises; G2 (*n* = 31): new footwear and heel raises; G3 (*n* = 31): usual footwear and orthoses; G4 (*n* = 31): new footwear and orthoses	10.89 ± 1.48	12	OxAFQ-C; FPS; WBL	G1 and G2 (using heel lifts) resulted in significant differences in OxAFQ for children’s scores in the physical domain over G3 and G4 (using orthoses) at the 1st and 2nd month measurements from baseline. No significant differences were found in the other measurements in OxAFQ, VAS scale pain, and WBL in the use of the two types of footwear comparing the use of orthoses or heel lifts. No differences in the groups were shown in OxAFQ scores comparing heel raise vs. prefabricated orthoses or comparing usual footwear vs. athletic footwear at 6 and 12 months from baseline. Significant differences were found (but inconsistently observed) regarding the footwear domain in OxAFQ. Children prefered using footwear, and parents prefered using athletic footwear.
Kuyucu et al. (2017) [32]	27 (0%)	G1 (*n* = 11): undergoing Kinesio taping; G2 (*n* = 11): mimicked Kinesio taping (sham);	13.18 ± 2.15	6	VAS; AOFAS scale; monitoring for recurrences; X-ray monitoring	Both groups experienced a significant reduction in pain versus at baseline. Both groups experienced a decrease in VAS score pain at 1st-week, 3rd-month, and 6th-month treatment versus baseline. VAS score pain showed no differences in the scores between groups after treatment. Both groups experienced a significant decrease in VAS scores in the 1st week, 1st month, 3rd month, and 6th month compared with the pre-treatment scores. Both groups experienced a significant increase in AOFAS scores at 1st-week, 3rd-month, and 6th-month treatment. G1 demonstrated significant improvements compared to G2 AOFAS scores at 1st-week and 3rd-month measurements.
Perhamre et al. (2011) [35]	50 (0%)	Pilot group (*n* = 5); G1 (*n* = 15): patients with heel cup; G2 (*n* = 15): patients who did not undergo intervention for pain; G3 (*n* = 15): control group	-	1 month	VAS; PA level with the Engstrom scale; radiographic examination—heel pad thickness	Children in symptomatic groups continued their high levels of PA during the study period. A significant increase in heel cup thickness between being barefoot, wearing shoes without a heel cup, and wearing shoes with the heel cup was observed. No differences in thickness were shown between using football sports shoes and indoor sports shoes. Maximal peak pressure was significantly reduced when comparing sport shoes with and without a heel cup, favoring the heel cup. G1 resulted in significant pain reduction in sport activities during the study.
Perhamre et al. (2011) [34]	35 (0%)	G1 (*n* = 18): heel wedge; G2 (*n* = 17): heel cup; G3 (*n* = 5): no intervention	-	2 months	Pain CR-10 (Borg); VAS; PA level with the Engstrom scale	G1 and G2 showed a significant decrease value in the VAS score level of pain after using a heel cup. The median pain level decreased in both activities (A, the more painful activity, and B, the less painful activity). Significantly decreased values were shown in the pain level of the B activity (less painful); G3 showed less pain after intervention (7 to 6), but there were no significant improvements.
Perhamre et al. (2011) [33]	51 (0%)	G1 (*n* = 20): heel cup; G2 (*n* = 24): heel wedge	-	6 months	Pain CR-10 (Borg); VAS; PA level with the Engstrom scale	Both groups no differences showed in median pain level and pain history at baseline. G1 showed significant differences in pain compared to G2. A total of 77% of subjects preferred to choose the heel cup in the third intervention period. Both groups showed less pain in all three intervention periods. Children continued their high levels of PA (Engstrom scale) during the study. Pain significantly decreased in groups A and B of PA. More PA resulted in less pain when using the heel cup and heel wedge.
Sweeney et al. (2023) [36]	43 (90.6%)	G1 (*n* = 16): Tuli’s Cheetah Heel Cup G2 (*n* = 16): Tuli’s The X Brace	10.3 ± 1.6	3	OxAFQ-C: physical, school or play, and emotional scores; VAS	G1 and G2 showed no differences in OxAFQ-C physical scores at the baseline or 3-month assessment. G1 and G2 showed no differences in OxAFQ-C school or play scores at baseline. G1 and G2 showed no differences in VAS scores during rest, during sports, or during activities of daily living at the baseline or 3-month assessment. G1 significantly improved the OxAFQ-C emotional scores versus G2 at the 3-month assessment. G1 and G2 showed a decrease in pain in OxAFQ-C scores at the 1-month, 2-month, and 3-month assessment from the baseline. No effects were shown in VAS scores at rest after 1 month, 2 months, and 3 months. Positive effects were observed in VAS scores for daily living activities and during sports in both groups at the 2- and 3-month assessments from the baseline.
Wiegerinck et al. (2016) [38]	101 (25%)	G1 (*n* = 32): wait and see; G2 (*n* = 33): heel raise (ViscoHeel, Bauerfeind); G3 (*n* = 33): physical therapy	10.63 ± 1.63	3	FPS-R (revised) with an FPK/FPN mechanic algometer; VAS and OxAFQ (children and parents)	No differences were observed between groups in FPS-R and OxAFQ scores at baseline. In all the treatments, pain significantly decreased after 3 months. Three groups showed a significant improvement in FPS-R and OxAFQ scores after treatment. G2 and G3 treatment showed significant benefits in OXaFQ scores for children and parents at 6 weeks, in comparison to the G1. No significant differences were shown between groups at the 3rd month in OxAFQ scores for children and parents. G3 showed differences in VAS score pain at 6 weeks compared to G1 and G2.

G1: Group 1; G2: Group 2; OxAFQ-C: Oxford Ankle Foot Questionnaire for Children; VAS: Visual Analogue Scale; AOFAS: American Orthopaedic Foot and Ankle Society (Ankle–Hindfoot Scale); FPS: Faces Pain Scale; WBL: Weightbearing lunge test; PA: Physical activity.

## Data Availability

The data presented in this study are available on request from the corresponding author.

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
