# Peer review of "Conservative Treatment of Sever’s Disease: A Systematic Review"

_jcm, 2024, doi:10.3390/jcm13051391_

Round 1

Reviewer 1 Report

Comments and Suggestions for Authors

Dear Authors,

I would like to express my gratitude for the opportunity to review this manuscript.

The manuscript at this stage requires improvements. Below are suggestions with line indication:

2 – Please revise the title considering the journal template and instructions for authors.

3-11 – Please revise the authors and affiliations format, considering the journal template and instructions for authors.

16-18 – Please include more information regarding data collection, for example the time associated with data collection.

29-58 – Please consider developing the introduction section. There is space for more text and references (in this version only 11).

75 – Please revise the text.

75-76 – Please format Table 1 considering the journal template and instructions for authors.

2.2 and 3.1 titles are the same, please consider reformulation.

109 “[17–24]” – Why are so many references associated with a methodological procedure? This is not clear.

109 – Please improve the figure quality.

Table 2 – Please revise the footnote format, considering the journal template and instructions for authors.

117 – Figure 2 content should be revised considering the journal template and instructions for authors. Additionally, the quality of the figure should be improved.

146 – Please revise the table content (all 4 pages), as well as the page headers incorrection regarding the number.

157-158 – Citations are incorrect. Please revise (and consider the same in other lines – 153-197 and all manuscript).

203-218 / 219-224 – Please consider standardizing the paragraph size to increase readability.

 199-261 - Please consider developing the discussion section. There is space for more text and particularly, more references.

265 – “Sever's disease” – It was already abbreviated in the first line of the introductions section. Please revise all manuscript.

262 - Please consider clear and take-home messages in this section

285 - Please carefully revise the references format, they are not according to the journal template.

Please consider improving the English in the document.

Comments on the Quality of English Language

Moderate editing of English language required.

Author Response

Dear Editor and Reviewers of Physiotherapy Theory and Practice:

Thank you very much for your suggestions and contributions to improve the quality of the manuscript. Following your indications, we respond, point by point, to the reviewers' comments.

In the text, all the modified or added sentences have been written in red to facilitate the correction by the reviewers.

  1. The manuscript at this stage requires improvements. Below are suggestions with line indication.

We extend our sincere appreciation for your constructive comments and favorable assessment of our research. We have taken your suggestion into consideration and have made a thorough revision of the discussion section in our manuscript.

  1. Please revise the title considering the journal template and instructions for authors.

The authors have proceeded to change the title in accordance with the journal's rules of style.

  1. Please revise the authors and affiliations format, considering the journal template and instructions for authors.

We have modified the affiliations according to what the journal template and instructions for authors.

  1. Please include more information regarding data collection, for example the time associated with data collection.

We have added the data collection.

  1. Please consider developing the introduction section. There is space for more text and references (in this version only 11).

The authors have expanded the introduction section and increased the number of bibliographical references. 

  1. Please revise the text.

The authors have reviewed the text of line 75.

  1. Please format Table 1 considering the journal template and instructions for authors.

We have modified the table according to what the journal template and instructions for authors.

  1. 2.2 and 3.1 titles are the same, please consider reformulation.

We have reformulated 3.1 title.

  1. 109 “[17–24]” – Why are so many references associated with a methodological procedure? This is not clear.

They refer to the 8 articles selected for the systematic review. To avoid confusion, we have decided to eliminate ourselves.

  1. 109 – Please improve the figure quality.

Following your recommendation, the authors have used the highest possible image quality. We have found that with 100% vision it does not generate any distortion in the image.

  1. Table 2 – Please revise the footnote format, considering the journal template and instructions for authors.

We have revised the footnote format.

  1. 117 – Figure 2 content should be revised considering the journal template and instructions for authors. Additionally, the quality of the figure should be improved.

The authors have adapted to the format of the magazine following the instructions for authors and have used the highest possible image quality. We have found that with 100% vision it does not generate any distortion in the image.

  1. 146 – Please revise the table content (all 4 pages), as well as the page headers incorrection regarding the number.

We have revised the table an also modified the order of the articles and change the page headers.

  1. 157-158 – Citations are incorrect. Please revise (and consider the same in other lines – 153-197 and all manuscript).

We have had an error with the bibliographic manager Zotero and it had not calculated the bibliographies correctly, it has already been modified and revised throughout the manuscript.

  1. 203-218 / 219-224 – Please consider standardizing the paragraph size to increase readability.

We have reviewed and standardized them.

  1. 199-261 - Please consider developing the discussion section. There is space for more text and particularly, more references.

Following your advice, the authors have expanded the discussion section and added new bibliographical references.

  1. 265 – “Sever's disease” – It was already abbreviated in the first line of the introductions section. Please revise all manuscript.

The authors have reviewed it throughout the manuscript. 

  1. 262 - Please consider clear and take-home messages in this section

We have modified it.

  1. 285 - Please carefully revise the references format, they are not according to the journal template.

We have checked the references of the document according to the journal template.

  1. Please consider improving the English in the document.

The authors have revised the English version of the manuscript with the help of a native translator.

Once again, thank you very much for the time spent and the interest shown in this work; as well as in the positive evaluations you have given of it.

Receive a warm greeting,                                                                            

The authors.

Reviewer 2 Report

Comments and Suggestions for Authors

Thank you for study.

 Conservative treatment of Sever's disease: a systematic review that was summarized the work done in general.

It would be useful for the authors to state a few points in the review.

It is known that the orthotic approach is used in the treatment of Severs diease. Heel lift, heel cup, insole and foot orthosis mentioned in the studies are all different orthotic approaches. More detailed information can be given about the orthotic approaches in the 8 articles examined in the study. Material used, thickness, custom made? Can the insole features be briefly mentioned?

Author Response

Dear Editor and Reviewers of Physiotherapy Theory and Practice:

Thank you very much for your suggestions and contributions to improve the quality of the manuscript. Following your indications, we respond, point by point, to the reviewers' comments.

In the text, all the modified or added sentences have been written in red to facilitate the correction by the reviewers.

  1. Comments to the Author: Conservative treatment of Sever's disease: a systematic review that was summarized the work done in general.

It would be useful for the authors to state a few points in the review.

It is known that the orthotic approach is used in the treatment of Severs diease. Heel lift, heel cup, insole and foot orthosis mentioned in the studies are all different orthotic approaches. More detailed information can be given about the orthotic approaches in the 8 articles examined in the study. Material used, thickness, custom made? Can the insole features be briefly mentioned?

We are grateful for your positive feedback and your suggestions for improving the quality of this work. In addition, we appreciate your positive vote for the acceptance of this project.

Following your valuable advice, the authors have added information on orthotic approaches between lines 225-231.

Once again, thank you very much for the time spent and the interest shown in this work; as well as in the positive evaluations you have given of it.

Receive a warm greeting,                                                                            

The authors.

Reviewer 3 Report

Comments and Suggestions for Authors

I think you have done an important review study on Sever's disease.

I have a few questions regarding this.

1. 63. Data analysis is important in system review. Please provide more details about the statistical techniques used to analyze this data.

2. 75. An explanation should be added for each of Table 1, P->, I->, C->, O->.

3. In the conclusion, please briefly add the male and female characteristics of Sever's disease.

Comments on the Quality of English Language

This paper needs some English editing by a reputable publisher.

Author Response

Dear Editor and Reviewers of Physiotherapy Theory and Practice:

Thank you very much for your suggestions and contributions to improve the quality of the manuscript. Following your indications, we respond, point by point, to the reviewers' comments.

In the text, all the modified or added sentences have been written in red to facilitate the correction by the reviewers.

  1. Comments to the AuthorI think you have done an important review study on Sever's disease. I have a few questions regarding this.

We are grateful for your positive feedback and your suggestions for improving the quality of this work.

  1. 63. Data analysis is important in system review. Please provide more details about the statistical techniques used to analyze this data.

The authors have added information on the analysis of the data.

  1. 75. An explanation should be added for each of Table 1, P->, I->, C->, O->.

Following your recommendation, the authors have added information in table 1.

  1. In the conclusion, please briefly add the male and female characteristics of Sever's disease.

The authors agree with his advice and have included this information in the conclusions.

Once again, thank you very much for the time spent and the interest shown in this work; as well as in the positive evaluations you have given of it.

Receive a warm greeting,                                                                            

The authors.

Reviewer 4 Report

Comments and Suggestions for Authors

The present review focused on conservative treatments for pain reduction of Sever's disease that still represents an important issue in rehabilitation. The review is well structurated with a good method and results section. The guidlines for systematic review were respected. 

I only suggested to improved the quolity of introduction and discussion sections. Please add more references in the introduction section that support your work and in the discussion section that support your main findings. 

Author Response

Dear Editor and Reviewers of Physiotherapy Theory and Practice:

Thank you very much for your suggestions and contributions to improve the quality of the manuscript. Following your indications, we respond, point by point, to the reviewers' comments.

In the text, all the modified or added sentences have been written in red to facilitate the correction by the reviewers.

  1. Comments to the Author: The present review focused on conservative treatments for pain reduction of Sever's disease that still represents an important issue in rehabilitation. The review is well structurated with a good method and results section. The guidlines for systematic review were respected. 

I only suggested to improved the quolity of introduction and discussion sections. Please add more references in the introduction section that support your work and in the discussion section that support your main findings. 

Thank you very much for your contribution, we appreciate your comments to improve our work, we have expanded the introduction and further developed the discussion. To this end, we have increased the number of references. Thank you again

Once again, thank you very much for the time spent and the interest shown in this work; as well as in the positive evaluations you have given of it.

Receive a warm greeting,                                                                            

The authors.

Round 2

Reviewer 1 Report

Comments and Suggestions for Authors

Dear Authors,

Thank you for considering my suggestions and incorporating them into the manuscript, which is globally improved, congratulations. Below are some specific suggestions with page indications:

Page 1 – Please revise the journal template, for example, the page header.

30-68 – Although the introduction section was expanded, there is still room for more text and citations aiming to introduce the topic to the readers.

132 - Please improve the quality of Figure 1.

132 – Please confirm if the Figure 2 text is according to the journal template (type and size of letter).

150 - 154 – The citations are incorrect, please revise in these lines and throughout the manuscript.

Table 2 – Please carefully revise the table content. For example, in study 2 and others, the end-point is missing in “et al.” and Kuyucu et al (2017)[24] age format is incorrect. All details should be carefully checked and corrected.

202-207 - Some paragraphs are too short, and others too long. Please consider standardization aiming to improve readability.

Please carefully revise the references format, they are not according to the journal template. Some examples, journals abbreviated and in full, some in uppercase, and others in lowercase.

The English improved, nevertheless, please revise aiming to improve details.

Comments on the Quality of English Language

Minor editing of English language.

Author Response

Dear Editor and Reviewers of Journal of Clinical Medicine:

Thank you very much for your suggestions and contributions to improve the quality of the manuscript. Following your indications, we respond, point by point, to the reviewers' comments.

In the text, all the modified or added sentences have been written in red to facilitate the correction by the reviewers.

  1. Thank you for considering my suggestions and incorporating them into the manuscript, which is globally improved, congratulations. Below are some specific suggestions with page indications:

Thank you very much for your suggestions and contributions to improve the quality of the manuscript.

  1. Page 1 – Please revise the journal template, for example, the page header.

The authors have proceeded to change the page header in accordance with the journal's style rules.

  1. 30-68 – Although the introduction section was expanded, there is still room for more text and citations aiming to introduce the topic to the readers.

Following your recommendation, the authors have expanded the introduction, adding important new information to bring the work up to date, and have increased the number of references from 11 in the first version to 26 in this latest version.

  1. 132 - Please improve the quality of Figure 1.

Following your recommendation, the authors have utilized the highest possible image quality. We have confirmed that at 100% zoom, the image does not exhibit any distortion.

  1. 132 – Please confirm if the Figure 2 text is according to the journal template (type and size of letter).

The authors have confirmed that the font type and size (Palatino Linotype 9) are correct, adhering to the journal's template, which corresponds to the preset style MDPI_5.1_figure.

  1. 150 - 154 – The citations are incorrect, please revise in these lines and throughout the manuscript.

The authors have made corrections to the citations throughout the manuscript.

  1. Table 2 – Please carefully revise the table content. For example, in study 2 and others, the end-point is missing in “et al.” and Kuyucu et al (2017)[24] age format is incorrect. All details should be carefully checked and corrected.

The authors have corrected and thoroughly reviewed all the details of the entire table.

  1. 202-207 - Some paragraphs are too short, and others too long. Please consider standardization aiming to improve readability.

The authors have standardized the paragraphs throughout the manuscript.

  1. Please carefully revise the references format, they are not according to the journal template. Some examples, journals abbreviated and in full, some in uppercase, and others in lowercase.

The authors have reviewed and revised the format of the references throughout the manuscript.

  1. The English improved, nevertheless, please revise aiming to improve details.

The authors have reviewed and revised the English language throughout the manuscript.

Once again, thank you very much for the time spent and the interest shown in this work; as well as in the positive evaluations you have given of it.

Receive a warm greeting,                                                                            

The authors.